# Cellulose Nanocrystals and Corn Zein Oxygen and Water Vapor Barrier Biocomposite Films

**DOI:** 10.3390/nano11010247

**Published:** 2021-01-18

**Authors:** Tal Ben Shalom, Shylee Belsey, Michael Chasnitsky, Oded Shoseyov

**Affiliations:** Robert H. Smith Faculty of Agriculture, Food and Environment, The Center for Nano Science and Nano Technology, The Hebrew University of Jerusalem, Rehovot 76100, Israel; tal.benshalom@mail.huji.ac.il (T.B.S.); shylee.belsey@mail.huji.ac.il (S.B.); Michael.Chasnitsky@mail.huji.ac.il (M.C.)

**Keywords:** cellulose nanocrystals, zein, flexible packaging, biocomposite, barrier films

## Abstract

Cellulose nanocrystals (CNC) are well-suited to the preparation of biocomposite films and packaging material due to its abundance, renewability, biodegradability, and favorable film-forming capacity. In this study, different CNC and corn zein (CZ) composite films were prepared by adding CZ to the CNC suspension prior to drying, in order to change internal structure of resulting films. Films were developed to examine their performance as an alternative water vapor and oxygen-barrier for flexible packaging industry. Water vapor permeability (WVP) and oxygen transmission rate (OTR) of the biocomposite films decreased significantly in a specific ratio between CNC and CZ combined with 1,2,3,4-butane tetracarboxylic acid (BTCA), a nontoxic cross linker. In addition to the improved barrier properties, the incorporation of CZ benefitted the flexibility and thermal stability of the CNC/CZ composite films. The toughness increased by 358%, and Young’s modulus decreased by 32% compared with the pristine CNC film. The maximum degradation temperature increased by 26 °C, compared with that of CNC film. These results can be attributed to the incorporation of a hydrophobic protein into the matrix creating hydrophobic interactions among the biocomposite components. SEM and AFM analysis indicated that CZ could significantly affect the CNC arrangement, and the film surface topography, due to the mechanical bundling and physical adsorption effect of CZ to CNC. The presented results indicate that CNC/CZ biocomposite films may find applications in packaging, and in multi-functionalization materials.

## 1. Introduction

Plastic packaging waste has recently become a major issue of interest. Several environmental pollution laws and regulations are indeed a consequence of the growing concern of governments over non degradable petroleum-derived plastics used for food packaging [1,2,3,4]. Hence many researchers and industrial companies have begun to focus more on renewable and biodegradable polymers instead of petroleum-derived plastics in order to comply to the new laws of the EU and other governments around the world [5].

There are three major types of bio-degradable components derived directly from biological material that can be used to produce bio-degradable nontoxic food packaging: polysaccharides, lipids, and proteins. In this paper, in order to reach the required features of excellent food packaging material, we chose to work with two of these components, polysaccharides and proteins [6,7,8,9].

The most popular polysaccharide on earth is cellulose, but there are many cellulose derivatives that can be used for food packaging [10]. In recent years many studies found crystal nanocellulose (CNC) in particular as unique materials that can be used for many different application Such as photonics, films and foams, surface modifications, nanocomposites, and medical devices [11,12,13,14].

CNC is one of the most suitable materials for enhancing properties of composite materials [15,16]. CNC, consistent with the required food packaging features especially strength and barrier properties, however, One notable disadvantage of CNC is its low resistance to moisture and in the deterioration in mechanical properties of films made from CNC when exposed to moisture [17,18,19,20].

Zein is a protein which belongs to the prolamin protein family that is produced as the non-water- soluble fraction of corn. It has been defined as an alcohol-soluble protein that occurs principally in protein bodies of maize endosperm [21,22]. Zein protein is used in the industry mostly due to its film-forming ability when cast from appropriate solvent systems. A film formed after solvent evaporation is tasteless, but it is glossy, scuff-resistant, and grease-resistant [23]. Zein sheets without plasticizer, however, resulted in a very brittle film that cannot be used for any packaging application [24,25,26]. Yet, an addition of plasticizers increases flexibility and produce much softer films [27]. Many studies that evaluate the effect of plasticizers on zein films for food packaging have already been done [28,29]. Nevertheless, a major disadvantage of using plasticizers is the theory that plasticizers are to decrease the intermolecular forces along polymer chains, imparting increased film flexibility while decreasing barrier abilities of film [30]. When chain mobility increases, the coefficient of diffusion also increases thus enhancing the gas and water vapor permeability; this significantly impedes the barrier properties and increases the packaging films water absorption capacity [30,31].

In this paper, CNC and CZ were successfully blended in different ratios to form a stable suspension. Casting method were developed, and green and nontoxic cross-linking method were also combined in the films forming process. 1,2,3,4-butanetetracarboxylic acid (BTCA) and Sodium hypophosphate (SHP) were added to the films to examine the effect of the crosslinker on the mechanical, optical, thermal and barrier properties of nanocomposite films. Carboxylic acids, particularly polycarboxylic acid, have been shown to be good cellulose crosslinking agents [32,33,34]. SHP is a most effective catalyst for catalyzing a reaction with BTCA [35] in a previous paper we demonstrate the effect of BTCA on CNC and CNC-poly vinyl alcohol film properties [16]. This nontoxic crosslinking method serves the textile industry in crosslinking cotton cellulose to improve anti-pilling, wrinkle recovery, antimicrobial, water repelling, and flame-retarding properties of the cotton fabric [36].

The non-crosslinked films exhibited excellent film flexibility but poor barrier ability and water absorption capacity, while crosslinked films showed good performance in both properties although the contact angel of the non-crosslinked films was higher, probably because of the unique arrangement of the films surface, as characterized by SEM and AFM. The objective of this study was to evaluate and characterize the films barrier, mechanical, optical and thermal properties of CNC/CZ based film as affected by the composition of “green” and nontoxic cross linker, a method developed in our previous publication [16].

This bio based composite material with its unique properties can meet the required features for an excellent food packaging material.

## 2. Materials and Methods

### 2.1. Materials

All chemicals, unless otherwise stated, were purchased from Sigma-Aldrich (St. Louis, MO, USA). A 3%wt suspension of CNC (sulfur content: 91 mmol kg^−1^, Zeta-potential: −28.5 mV) was produced from cellulose pulp according to already established method of sulfuric acid hydrolysis, [26] and kindly supplied by Melodea Ltd. (Rehovot, Israel).

### 2.2. Blending and Stability Tests

CNC particles size was evaluated by transmission electron microscopy (TEM) (FEI Tecnai G12). TEM images of the aqueous CNC suspension revealed rod-like particles (length: 20–500 nm [mean 170 nm]; width: 2–20 nm [mean 6.8 nm]). A clear and transparent CNC suspension (2 wt%) that was prepared by diluting the 3% suspension provided in distilled water, was used in this work.

Pure zein powder was dissolved in a solution of 1:4 wt. ratio of distilled water:ethanol to obtain a 15 wt.% concentrated CZ solution. The zein solution was then dispersed, at various ratios, in 15 mL of 2% wt. suspension of CNC or CNC/BTCA/Sodium hypophosphate (SHP) for crosslinked films (see Table 1 1,2,3,4-Butanetetracarboxylic acid (BTCA) powder (10 mM) and sodium hypophosphite (SHP) monohydrate (5 mM) were added to the suspensions of the cross-linked samples prior to sonication of the suspension as describe in our previous publication [16]). The mixtures were sonicated at 220 W for 3 min before being cast.

Flocculation was experimentally determined using the “jar test,” all the samples suspensions settling for 5 days at room temperature.

### 2.3. Self-Standing Films Fabrication

The suspensions were gently mixed and degassed, 15 mL CNC/CZ or CNC/CZ/BTCA/SHP suspensions were cast onto a polystyrene petri dish and allow to dry for several days under ambient conditions until constant weight was achieved. Yielding 50 ± 5 µm thick films. Finally, all samples were subjected to a 3-day conditioning period, under ambient conditions (23 ± 2 °C and 60 ± 1% relative humidity).

### 2.4. Optical Transmittance Test

Optical transmittance of 30 mm-long × 20 mm-wide and 40 ± 10 µm-thick film samples was determined with a UV-Visible spectrophotometer (Thermo Scientific Evolution 220), in visible light (220–1100 nm), at room temperature. The transmittance spectra were acquired using air as blank. The spectra were analyzed by Thermo Scientific Insight 2.

### 2.5. Water Vapor Permeability (WVP)

WVP of each type of film was determined based on method as described by the American Society of Testing Materials (ASTM) standard method E96-66, using a coulometric sensor device by Mocon AQUATRAN 3/38 WVTR Analyzer.

### 2.6. Oxygen Permeability

Oxygen Transmission Rate (OTR) measurements have been carried out using a coulometric sensor device by Mocon Oxtran 2\22 at a pressure of 1 atm, a temperature of 23 °C and 50% of relative humidity in accordance with the ASTM D3985-05 standard. All samples are coated on a 30 µm corona treated biaxially-oriented polypropylene (BOPP) film since the device did not allow for the insertion of the self-standing film by itself with the aim of simulating food packaging material. Coating of BOPP surfaces were achieved using an Elcometer 4340 motorized film applicator. A range of applicator head attachments can be used defining the thickness of the coating, allowing the user to select the most appropriate for their specific use, in this study 80 μm spiral bar are used. Yielding 1.5 µm thick coating.

### 2.7. Water Contact Angle

The contact angle of water droplets on the surface of the films were measured using sessile drop on a contact angle goniometer (Rame-hart, INC. 100-00). Samples were cut into rectangles (2 cm × 2 cm) before using a micro syringe to drop a consistent amount of water on the film samples. The contact angle was manually measured after the drop had settled for 4–5 s and a constant contact angle has been achieved.

### 2.8. Water Absorption Test

The rate of water absorption was determined in accordance with ASTM: D570. Film specimens were dried in an oven at 110 °C, for 3 h, and then immediately placed in a desiccator to cool. After cooling, the specimens were weighed and then submerged in water, at 23 °C, for 4 days. Every 24 h, the specimens were removed, patted dry with a lint-free cloth, weighed and re-submerged in water.

Films water absorption was determined using Equation (1):%Water absorption = [(Wet weight − Dry weight)/Dry weight] × 100(1)

### 2.9. Scanning Electron Microscopy (SEM)

Films surface and cross-sections were examined. The dry films samples were immersed in liquid nitrogen and fractured. Films were sputter-coated with gold (12 mA, 6 min) using an S150 sputter coater (Edwards, UK), and imaged with a MagellanTM 400 L scanning electron microscope.

### 2.10. Atomic Force Microscope (AFM)

AFM analysis were performed using an JPK Nanowizard4 (JPK BioAFM Bruker Nano GmbH, Germany), with qp-BioAC probes, the 80-um long cantilever (NANOSENSORS, Switzerland), in AC (tapping) mode at a 1.6 Hz line scan rate m/s^−1^ scan speed. The data were analyzed by Gwyddion software.

### 2.11. Mechanical Testing

Mechanical tensile tests of CNC/CZ or CNC/CZ/BTCA/SHP nanocomposite films were performed using an Instron 3345 Tester, (2 mm min^−1^), equipped with a 100 Newton load cell. Stress-strain curves were plotted, and Young’s modulus was determined from the slope of the low strain region between 0.2−2% and elongation of the film. Toughness is defined as the energy of mechanical deformation pre unit of volume up until fracture and is calculated by the area under the entire stress-strain curve up until fracture. It is the maximum energy per unit volume that the sample can absorb before it fails completely.

### 2.12. Thermogravimetric Analysis (TGA)

The thermal properties of the CNC/CZ or CNC/CZ/BTCA/SHP nanocomposite films were measured using a thermogravimetric analyzer (TGA-Q500 TA Instruments). Samples of 10–20 mg were weighed and heated from 30–500 °C, at a rate of 10 °C/min. The nitrogen flow rate was 60 mL/min. Data was collected at 0.35 s/point.

## 3. Results and Discussion

### 3.1. Blending and Stability Tests

CNC/CZ nanocomposite films were prepared using a solution-casting technique. The aqueous CNC suspensions were cast onto polystyrene (90 mm diameter) Petri dishes. They were then allowed to stand under ambient conditions to allow the film to completely dry. BTCA and SHP were added to the films to examine the effect of the crosslinker on the mechanical, optical, thermal and barrier properties of nanocomposite films.

Theoretically, zein-protein is insoluble in water; however, it is not a totally hydrophobic polymer [37]. While in pure ethanol or water the protein cannot fully dissolve, there is a range of ethanol and water ratios in which the CZ protein is totally soluble; a ratio of 1:4 water:ethanol is the optimal. (Figure 1) Furthermore, a mixture of CNC/CZ yields a better suspension, with higher stability and improved results in stability test as seen in Figure 2.

This phenomenon can be explained by the hydrophobic interactions between CNC and CZ. Hydrophobic interactions are an important aspect of CNC/CZ film formation. CNC provided good links through hydrophobic interaction between CZ protein molecules [38]. This also indicates that BTCA as a non-polar molecule might only play a secondary role in forming a uniform film matrix.

### 3.2. Optical Transmittance Test

Transmittance spectra analyses of pristine CNC and CNC/CZ nanocomposite films, both crosslinked and not crosslinked, were performed in order to assess the film uniformity as well as transparency. Nanocomposite film transparency is a function of both CZ concentration and crosslinker content. In summary, the addition of the BTCA cross linker to nanocomposite CNC-CZ films generated films with improved transmittance at 500 nm for all the films other than films with ratio of 10:1 and 30:1. This could indicate a uniform film formation and chemical linkage between the components in the present nanocomposite films. The light transmittance rate decreased to 45.5%, 3.1%, for the nanocomposite films containing high amount of CZ,1:6 and 1:3 CNC-CZ ratio, respectively. However, the cross-linked 6:1 CNC-CZ films yielded improved transmittance of 54.3% and the cross-linked 3:1 CNC-CZ films yielded improved transmittance of 28.8% compared to the noncrosslinked 3:1 CNC-CZ films (Figure 3).

### 3.3. Water Vapor Permeability and Water Absorption of the Films

The water resistance of CNC/CZ films is characterized by a water vapor permeability (WVP) test and by water absorption tests, (Figure 4 and Figure 5). Figure 4b shows the water absorption over time; the absorption is drastically reduced in the crosslinked films suggesting that the tighter structures do not allow as much water in the films as their non-crosslinked counterparts. The non-crosslinked films disintegrated during the test and were not measured during the entire test period (Figure 4a). Xia et al. propose that combine crosslinking agents prevented the protein matrix from swelling in moist conditions [39]. Increasing ratios of CZ to CNC in the composite films also affect the water vapor permeability (WVP) of the film, and the addition of the BTCA cross linker to nanocomposite CNC-CZ films generated films with significantly improved WVP (Figure 5). Although the presence of CZ at the non-crosslinked films at ratio of 30:1, 10:1 and 6:1 yielded better WVP values, the crosslinked films yielded extremely low values of WVP and water absorption. Even though BTCA might have played a secondary role in film matrix in terms of mechanical and thermal properties, BTCA is important in preventing water absorption (Figure 4b). We suggest that the hydrophilic CZ may have filled pores and gaps in the films structure while BTCA generates chemical linkage between CNC particles and between CNC and CZ, based on their functional group. This proposed mechanism prevented the film from swelling and absorbing water and significantly improved WVP values (Figure 4 and Figure 5). CNC by nature is more hydrophilic than CZ, thus WVPs of films containing high amount of CZ were expected to be lower than those with low amount of CZ. However, in films containing a high concentration of CZ (6:1 and 3:1), we noticed that the WVP increased, probably due to the phenomenon that high concentration of CZ impairs the internal structure of the composite thereby allowing the passage of water vapor. The control CNC films fully disintegrated after a 12 h incubation in water, while the crosslinked CNC films absorbed, on average, 16.1 ± 1% water and were stable throughout the test period. The water absorption capacity of the CNC/CZ nanocomposite films was lower than the control film in the both the crosslinked and non-crosslinked series. The water absorption is lowered as the CZ concentration rises until it is peaked at 1:10. At ratios of 1:6 and 1:3 the water absorbance rose almost to the levels of the pure CNC films in both series (Figure 4).The lowest WVP value of, 2.03 ± 2.8 [g/m²·day] and the lowest water absorption capacity value of 4.4 ±0.8% were noted with crosslinked 15:1 CNC/CZ, suggesting that in this unique ratio between the CNC and CZ optimal interactions are obtained. We hypothesize that a higher concentration of CZ interferes with CNC self-assembly arrangement and impairs the film’s barrier ability.

### 3.4. Contact Angles

The surface properties of the films were investigated by contact angle tests, as shown in Figure 6. The water contact angle of non-crosslinked CNC films increased from 45.1 ± 1° to 62.6 ± 0.5° and 60 ± 0.5° in 15:1 and 10:1 CNC/CZ films, respectively, whereas the water contact angle of cross-linked CNC films only increased from 44.3 ± 0.5° to 59.8 ± 0.7° and 58.8 ± 1° in 15:1 and 10:1 CNC/CZ films respectively. The contact angle of water droplets of the films can be a way to measure the relative hydrophilicity of the films. The lower the contact angle is, the higher the ability of the water to wet the film as the surface tension of the water plays a role in the balance of free energy of the contact surface of the films. The addition of the relatively more hydrophobic CZ protein increases the contact angle of both the non-crosslinked and crosslinked CNC/CZ film in the 15:1 and 10:1 CNC/CZ films suggesting that unique hydrophobic interaction form in specific ratios between the CNC and CZ are the major enhancers in surface hydrophobicity and the water permeability ability through the films.

The addition of the relatively more hydrophobic CZ protein to the films increases the contact angle of the films, most prominently shown in the 15:1 and 10:1 CNC/CZ films. This suggests, like in the water absorption tests, that the CZ protein in these ratios causes a different internal structure to the particles in the films which makes them less water absorbent and results in a higher contact angle. Unlike the water absorption tests, the adding of the crosslinking agent did not show any significant difference in contact angle. This might suggest that the CZ protein changes how the CNC arrange themselves in the surface of the film as is shown in Figure 8. The higher contact angle at the non-crosslinked films is inconsistent with our proposed hypothesis and the results we obtained (Figure 4 and Figure 5), thus leading us to propose that unique structures form over the film surface.

### 3.5. Oxygen-Barrier Properties of Coated Films

Oxygen transfer rate (OTR) is a particularly important test when considering a certain material for food packaging. OTR plays a major role in the shelf life of food products as well as WVP. The OTR of the CNC/CZ coated BOPP films was measured at 23 °C and 50% RH (Figure 7) as a function of CNC/CZ ratio and cross linker. The OTR values of BOPP films has significantly decreased when CNC/CZ coating was applied.

As Figure 7 indicates, the application of CNC/CZ or cross linked CNC/CZ on BOPP films showed nearly more than two or three orders of magnitude in reducing OTR of the coated BOPP films when compared to BOPP alone; the OTR of BOPP film was determined as 2000 [cc/m²·day] [39]. It was reported that oxygen permeability values of zein films are lower than those of common synthetic plastic films [8,40]. We found that OTR value decreases further when adding CZ to the CNC suspensions, a dry CNC coating of 1.5 micron on a BOPP sheet reduced the OTR from 2000 [cc/m²·day] to ~176 [cc/m²·day] at 50% relative humidity. Using CZ as an additive resulted in a decrease in the OTR value of the coated BOPP to 11 [cc/m²·day] for the 15:1 CNC/CZ ratio and 12 [cc/m²·day] for the 10:1 CNC/CZ ratio for the crosslinked films, showing that the crosslinked formulation of CNC/CZ reduced the OTR value even further. The increase in OTR with CNC concentration is associated with the hydrophilic nature of CNC. The further breakdown of hydrogen bonds created additional sites for the dissolution of oxygen and increased the mobility of the oxygen molecules within the coating layer [39]. High humidity may also cause swelling of the film and allow easier and faster oxygen diffusion, this phenomenon is prevented by adding BTCA as a crosslinker. Results show that CZ and BTCA as a crosslinker agent can reduce OTR of CNC and improve the OTR value in humid conditions.

### 3.6. Film Morphology and Surface Characterization

The structural and morphological features of the self-assembled CNC films, crosslinked CNC and CNC/CZ nanocomposite films were also analyzed and characterized using SEM and AFM. The free surface SEM images (Figure 8) of CNC’s self-assembled films exhibited a relatively smooth surface. In addition, the high magnification SEM images of the surface revealed that the prepared self-assembled CNC films possessed a well-distributed nano-porous structure, while the crosslinked CNC and crosslinked CNC/CZ nano composite films demonstrated more aligned CNC assembly and flat fibrous structure at the nano-scale level (Figure 9). In contrast, the non-crosslinked CNC and crosslinked CNC/CZ nano composite films demonstrated “sheaf of wheat” like structure, this spectacular structure (Figure 8), is consistent with our hypothesis explaining the enhancement of the contact angle in the non-crosslinked films due to a unique surface structure. We suggest that CNC-CZ interactions led to unique topography causing a lotus-like effect that reduces the surface contact area of water droplet with the film surface which in turn leads to an improvement in the contact angle value.

The prepared films were characterized by AFM, deflection of the cantilever in AFM observation measurement over x and y coordinates used to determine the average surface roughness (R_a_) and root mean square roughness (R_q_). The surface roughness of films can be clearly seen in the AFM images, which is correlated with the SEM analysis. In addition, it was noted that CNC particles self-assembled in an orderly arrangement in films with CZ. The arrangement of CNC particles would result in the formation of needle- like shape structure that promotes compact conformation on the membrane surface, leaving a significant low nanoscale surface roughness of 3.8 nm.

Surface roughness of CNC films was changed depending on the CZ content and crosslinking. (Figure 10) Surface roughness of CNC films increased with an increase in CZ content up to 15:1 ratio of CNC: CZ; incorporation then leveled off in 10:1 ratio and sharply increased again at 6:1 and 3:1 ratio which is probably attributed to agglomeration of CZ that is embedded in the matrix of the film. Evidently, the increase in surface roughness of the nanocomposite films with high content of CZ indicates that high amount of CZ in CNC matrix can produce a non- homogeneous dispersion that can result poor film properties when dried. In the case of the crosslinked films (Figure 10b), a moderate increase in surface roughness can be seen, which is probably attributed to prevention of CZ agglomeration in CNC matrix even at high CZ concentrations; this phenomenon is positively correlated with the SEM analysis and transmittance tests.

### 3.7. Mechanical Properties

Mechanical properties of CNC, cross linked CNC, and CNC/CZ nanocomposite films with various CZ content were achieved by testing the stress strain curve for all film samples and are shown in Figure 10. The main factor to influence the mechanical properties is the amount of CZ added and whether it is crosslinked. The UTS of pure CNC film increased from 64.7 ± 6.5 MPa up to 73 ± 7.5 MPa with the addition of CZ in ratio of 15:1 for the crosslinked CNC films, the UTS increased from 66.9 ± 2.2 MPa of the cross-linked CNC film up to 82.6 ± 10.2 MPa with the addition of CZ in ratio of 30:1. A similar behavior of increase in UTS with an increase in the CZ content has been observed in various biopolymer-based composite films [41]. This behavior may be attributed to the CZ protein structure, but also by the stronger interfacial interaction through chemical bonds between CNC matrix and the protein interfacial area.

Additionally, rigidity of the CNC/CZ films, as determined in Young’s modulus increased with an increase in CZ content up to 10:1 ratio. The Young’s modulus increased from 3880.9 ± 517 to 7727.5 ± 198.7 MPa in pure CNC film and CNC/CZ 10:1, respectively. However, the change in the Young’s modulus of cross-linked films was not so distinctive; they were changed in an insignificant manner. However, compared with the non-crosslinked films, a significant change was observed. This reduction in Young’s modulus is affecting the film’s tensile strain, which is significantly higher in the crosslinked films except in CNC/CZ 6:1 and 3:1 ratios probably because dispersion difficulty and an interference with CNC self-assembly arrangement. Furthermore, the effective amount of CNC in the film has decreased, weakening the strength of the film.

The addition of a crosslinker yielded tougher films, the most marked improvement in toughness was observed in the crosslinked CNC\CZ film, more specifically, 30:1 CNC\CZ composite films showed a film toughness of 3.9 MJ/m³ (±0.7), a 358% increase as compared with the pristine CNC film (Figure 11).

### 3.8. Thermogravimetric Analysis

Thermogravimetric analysis (TGA) was carried out to investigate the thermal degradation of CNC and CNC/CZ composite films (Figure 12). TGA thermograms showed that weight loss began from 50 °C to 150 °C for the CNC and CNC/CZ composite films; all the films exhibited multiple steps of thermal degradation. Initial weight loss corresponded to the removal of moisture from the films, which existed in H-bonded form with the hydroxyl groups of the glucosyl units along CNC [42].

The second decomposition began between 200–250 °C, this can be attributed to the degradation of CNC. The major thermal decomposition was observed around 250–360 °C. The decomposition temperature increased due to the high CZ content, this phenomenon is associated with the protein degradation, suggesting that the CZ had stabilized the CNC to some extent.

It is believed that the temperature at which the polymer starts to degrade is the most important criteria to evaluate the thermal stability of polymer Incorporation of CZ resulted in slightly improved thermal stability. The results show that the temperature of highest rate of thermal degradation rises as the concentration of CZ rises. (Table 2) The higher thermal degradation temperature and better thermal stability of the composite were observed in the 6:1 CNC/CZ ratio of the crosslinked films, this can be attributed to the strong interaction between the CZ and the CNC matrix phase. (Figure 12) When BTCA crosslinker was added to the biopolymer, it seems that the cross linker had a stabilizing effect of the CNC films since the degradation temperature rises with crosslinking. This could suggest that the CZ content influenced the thermal stability of CNC films. Moreover, the optimal ratio between CNC and CZ could provide highly dispersed and distributed CZ layers within CNC matrix which led to improved thermal stability.

## 4. Conclusions

In this study, homogenous CNC/CZ bio-composite films were prepared by incorporating CZ to CNC suspensions at different ratios, in order to develop a novel film that meets the desired requirements of food packaging. Using easily available, cheap and degradable bio-materials in food packaging could prove very useful when considering the environmental consequences of current packaging waste. The mechanical, thermal, optical and the sealing capabilities of these films were tested in this study. The CNC/CZ films showed better flexibility when compared with a pure CNC film. The transparency of the films was slightly reduced as the concentration of the CZ proteins increases. When considering the water absorption capabilities of the films, there seems to be a range of CZ/CNC ratios that perform better. The ratios of 1:15 and 1:10 absorbed the least amount of water out of all the films, at higher concentrations of CZ there is an increase of water intake in the films suggests that the hydrophobic nature of CZ is not the only factor in play. In Surface images of SEM and AFM analysis suggest that in the films with ratios of 1:15 and 1:10 there is a bunching of the CNC rods that is similar to a “sheaf of wheat”. This unique structure can be caused by interactions between the CNC rods and the CZ protein and could likely be the cause of the decrease in water retention. This is further supported by the contact angle measurements that show that at these ratios the contact angle rises significantly in both crosslinked and not crosslinked films, further confirming that the combination of CNC/CZ causes changes in both the surface and bulk of the films. The decrease in the water absorption rate as well as the increase in the water contact angle indicated the hydrophobic nature of the film probably because the hydrophobicity of CZ and the unique interaction between the CNC and CZ, furthermore SEM and AFM analysis revealed “sheaf of wheat” like structure, this spectacular structure consistent with our hypothesis explaining the enhancement of the contact angle and in hydrophobicity due to a unique surface structure. Water vapor (WVP) and oxygen transmission rate (OTR) films significantly decreased by CZ incorporation to CNC matrix depending on CNC and CZ ratio and cross linking. The improvement of water vapor barrier properties and water absorption of CNC/CZ cross linked films were obtained with the 15:1 and 10:1 samples. The coating in these ratios improved the OTR of BOPP base film by showing nearly three orders of magnitude lower oxygen permeability than uncoated BOPP film and function as proper coating formulations.

The bio-composite films also showed improved thermal stability and the composite films increased the degradation temperature by 26 °C, compared with that of CNC film. These results may be attributed to the strong interactions of CNC rods and CZ protein and high degree of dispersibility of CZ within the CNC matrix.

In conclusion, CNC/CZ films could conceivably constitute an alternative to synthetic materials. Using CNC as a template to disperse various nanoparticles has significant potential in preparing multifunctional composite materials and coatings with barrier and humidity resistance for tomorrow’s materials.

## Figures and Tables

**Figure 1 nanomaterials-11-00247-f001:**
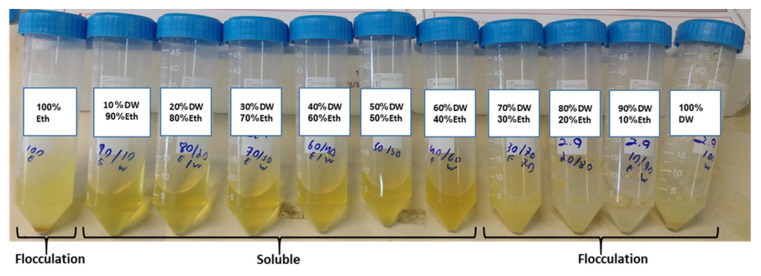
Solubility of CZ protein in ethanol /water mixture.

**Figure 2 nanomaterials-11-00247-f002:**
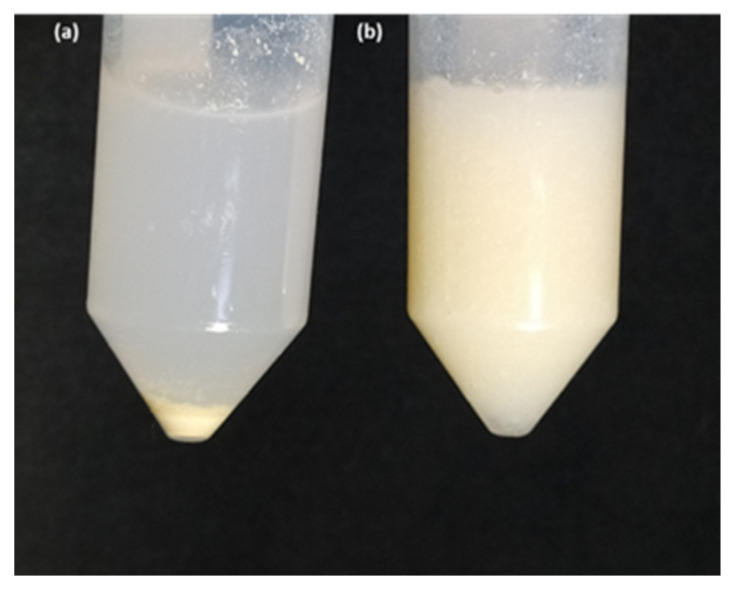
Suspension stability. (**a**) Flocculation of CZ in DW, (**b**) CNC/CZ improved suspension stability.

**Figure 3 nanomaterials-11-00247-f003:**
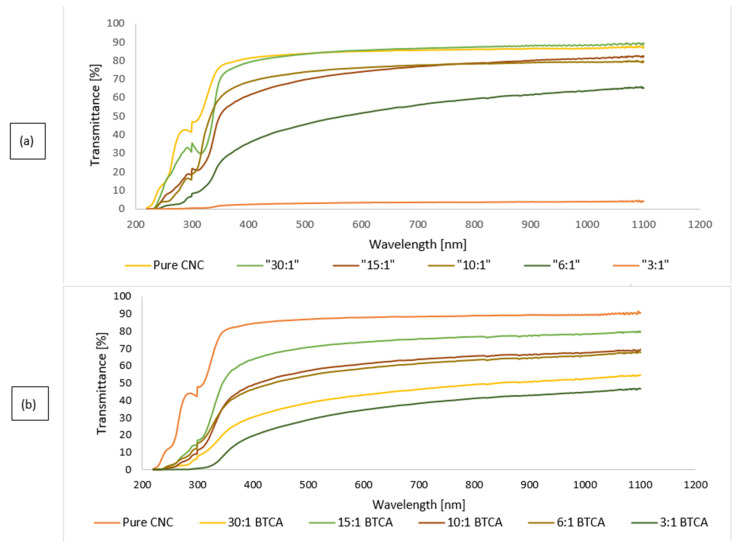
Transparency of CNC, crosslinked CNC films, and nanocomposite CNC/CZ films with different CZ contents. (**a**) Non crosslinked films, (**b**) crosslinked films.

**Figure 4 nanomaterials-11-00247-f004:**
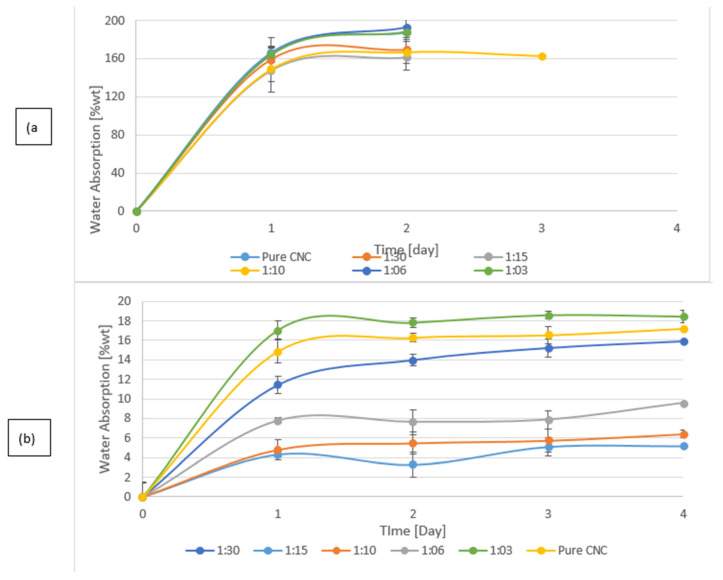
Water absorption over time. (**a**) non-crosslinked CNC and nanocomposite CNC/CZ films and (**b**) crosslinked nanocomposite CNC/CZ films.

**Figure 5 nanomaterials-11-00247-f005:**
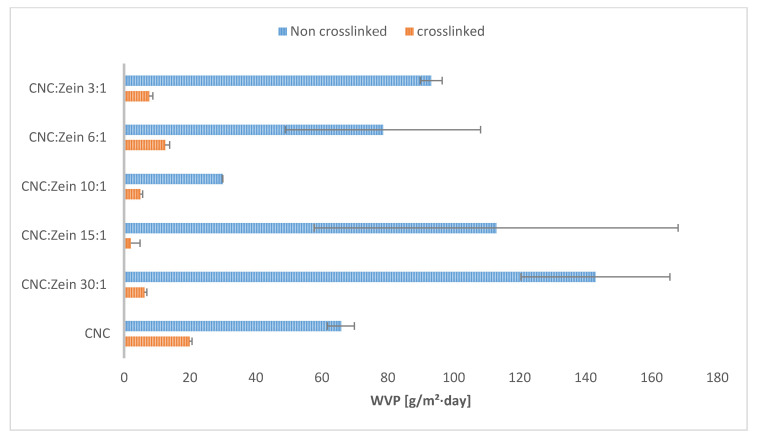
WVP of nanocomposite CNC/CZ film as affected by different CZ concentration and crosslinking.

**Figure 6 nanomaterials-11-00247-f006:**
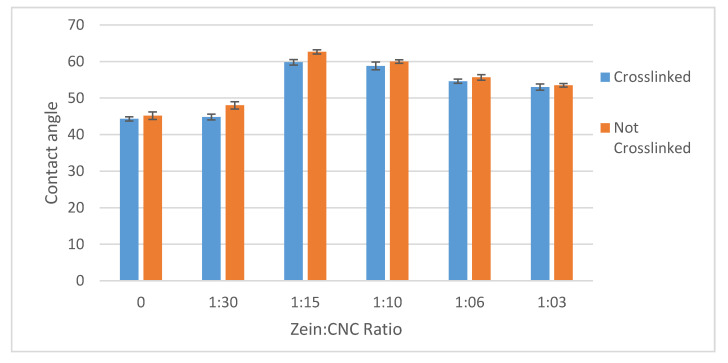
Films water contact angle as affected by different CNC/CZ ratio and crosslinking.

**Figure 7 nanomaterials-11-00247-f007:**
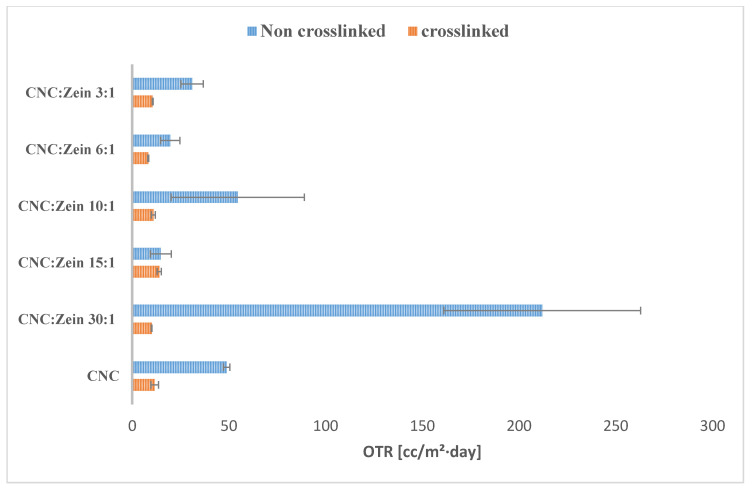
Comparative OTR of various films samples. All samples are coated on a 30 µm corona treated biaxially-oriented polypropylene (BOPP) film.

**Figure 8 nanomaterials-11-00247-f008:**
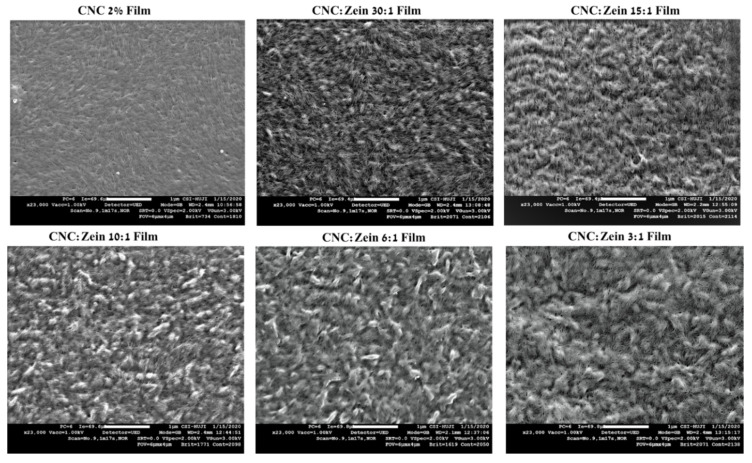
SEM images of non-cross linked CNC and CNC/CZ films. The images are at a magnification of 23,000×.

**Figure 9 nanomaterials-11-00247-f009:**
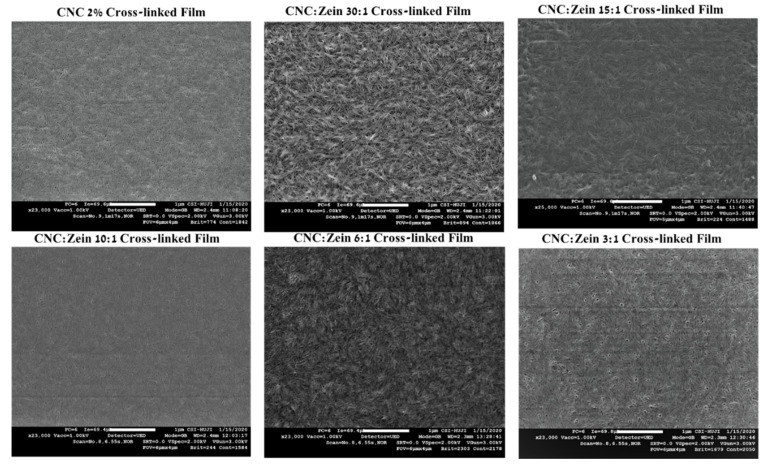
SEM images of cross linked CNC and CNC/CZ films. The images are at a magnification of 23,000×.

**Figure 10 nanomaterials-11-00247-f010:**
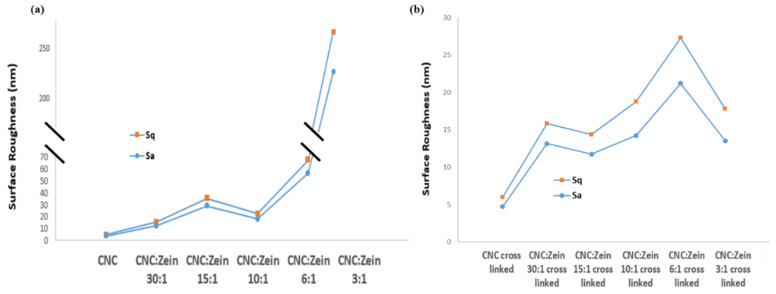
Effect of CZ content on surface roughness of CNC nanocomposite films. (**a**) Non-cross linked films (**b**) cross linked films.

**Figure 11 nanomaterials-11-00247-f011:**
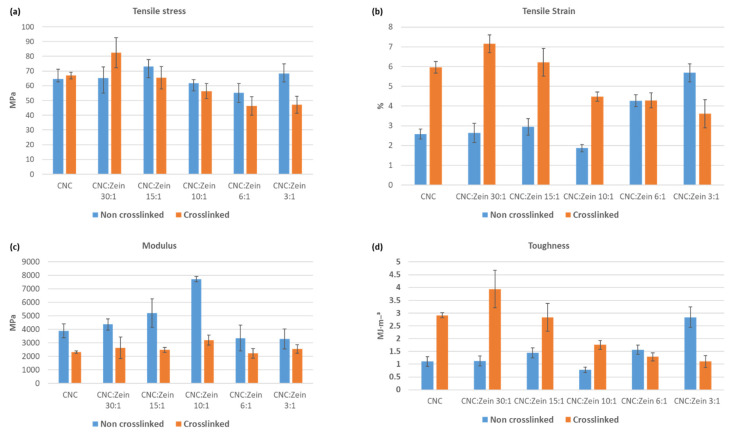
Tensile testing of crosslinked and non-crosslinked CNC films, nanocomposite CNC\CZ films (**a**) Tensile stress at break, (**b**) tensile strain at break, (**c**) Young’s modulus and (**d**) toughness.

**Figure 12 nanomaterials-11-00247-f012:**
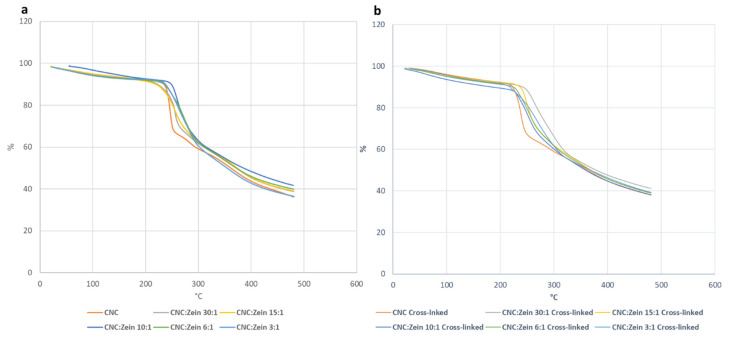
TGA curves of CNC and CNC/CZ films. (**a**) Non-cross linked films (**b**) cross linked films.

**Table 1 nanomaterials-11-00247-t001:** Compositions of CNC and CZ biocomposite films.

Sample	CNC:CZ Ratio [wt:wt]	CNC [wt%]	CZ [wt%]
CNC Control	-	2	-
CNC-Zein	30:1	1.935	0.065
CNC-Zein	15:1	1.875	0.125
CNC-Zein	10:1	1.818	0.182
CNC-Zein	6:1	1.714	0.286
CNC-Zein	3:1	1.5	0.5

**Table 2 nanomaterials-11-00247-t002:** Temperature at maximum rate of mass loss in the first degradation step for crosslinked and non-crosslinked films.

Crosslinked					
Pure CNC	1:30	1:15	1:10	1:06	1:03
245.69	259.64	259.3	260.28	271.55	262.27
**Non-Crosslinked**					
Pure CNC	1:30	1:15	1:10	1:06	1:03
237.73	242.35	250.34	251.01	251.69	260.35

## Data Availability

Dara is available on the request from the corresponding author.

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
