# Peer review of "Cellulose Nanocrystals and Corn Zein Oxygen and Water Vapor Barrier Biocomposite Films"

_nanomaterials, 2021, doi:10.3390/nano11010247_

Round 1

Reviewer 1 Report

The analysesd manuscript falls into the category of those who discuss new materials and functionalized surfaces, with applications in the field of packaging.

The material presented is of quality, with an important number of experimental determinations and tests that prove the qualities and properties of cellulose nano-crystals with corn zein biofilms.

There are some observations and suggestions to make:

  1. The entire material must be corrected from the point of view of inserting bibliographic references in the text. Pay attention to the Instructions for authors.
  2. Please improve the Introduction section with more details about the materials chosen.
  3. he acronyms used for the first time into the bulk text should be all explained, not only into the Abstract section (for example at “... cellulose and CNC ...” into the third paragraph on Introduction Section).
  4. Please improve the resolution of the figures and uniformed its (Eg. fig. 5, 6).

Author Response

Thank you very much for your comments.

Reviewer 2 Report

Dear Editor, Dear Authors

The manuscript titled Cellulose Nanocrystals and Corn Zein Oxygen and Water Vapor Barrier Biocomposite Films’ demonstrates interesting technical approach to combine two bio-derived ingredients with hierarchical structure in cast flexible films with improved barrier properties. The application of crosslinking system allowed manufacturing of CNC-based films that do not undergo disintegration in water and the introduction of hydrophobic corn zein reduced water permeability of CNC-based films.

From the point of view of material development and material engineering the concept of the work and observed effects are novel and interesting. Unfortunately, the quality of results presentation in the graphs, results description and discussion is of moderate to low quality - looks as it was done in rush. The high number of editing errors is distracting and the manuscript requires a lot of editing work. In the current form the work can not be published and I recommend a major review with resubmission for the second review.

General remarks, since the work presents two series of composite films, crosslinked and un-crosslinked, it would help the Readers to follow the results if there was a clear indication in the text which series is described and perhaps more clear separation of the discussion part focused on un-cross linked materials from that cross linked.

Detailed comments:

  1. Line 13: ‘incorporating CZ to CNC matrix’

We consider composite matrix as a continuous phase in which reinforcing filler is dispersed. In this case, CZ is not incorporated into the interior of cellulose crystals (dispersed in CNC), but is adsorbed on the surface of nanocrystals so it should rather be considered a surface modifier. In the applied range of contents and characteristics of ingredients it is difficult and not necessarily to indicate a matrix of the prepared composite films. They consist of two ingredients with hierarchical morphology.

  1. Line 21: ‘ These results can be attributed to the hydrophobic interaction’

What is the proof that the observed effects were because of hydrophobic interactions and not hydrophilic or chemical bonding?

  1. Line 22: ‘AFM analysis indicated that CZ could significantly affect the CNC arrangement’

Authors refer to AFM scans to illustrate the changes in CNC arrangements. The original AFM topography and phase scans should be incorporated into the manuscript or supplementary materials.

  1. Line 35: ‘There are three major types of bio-degradable components’

The references cited by Authors mention more types of bio-degradable components, including synthetic PLA and bacteria derived PHA. Perhaps Authors mean three natural components directly extracted from biomass.

  1. Line 45: ‘disadvantage of CNC is its resistance to moisture’

Do Authors mean 'low' resistance to moisture?

  1. Line 60: ‘blended in different ratios with an extremely low amount of solvent’

What amount of solvent is considered "extremely low"?

  1. Line 68: ‘a method developed in our previous publication.(16,32)’

Non of the Authors’ names from the cited article under the number 32 appears in the current manuscript. Pease check the reference numbering/citation.

  1. Line 74: What was the form of CNC - dry or suspension, if suspension what was the CNC concentration?
  2. Line 80: ‘TEM images of the aqueous CNC suspension’

What was the TEM conducted by the Authors of supplied by the producer? How actually the samples were prepared for analysis?

  1. Line 81: ‘CNC suspension (2 wt%) in distilled water, was used in this work’ .

Was it prepared by the Authors or supplied in that form by manufacturer?

  1. Line 84: ‘The aqueous zein solution’

I suggest to remove the word 'aqueous' since it is a solution in mixed ethanol and water solvent system.

  1. Line 85: ‘CNC/ BTCA/SHP’

SHP appears for the first time in the text. The abbreviation should be explained at a first use. A list of all chemicals used in this work with their abbreviations should be given in 'Materials' section for clarity.

  1. Line 86: ‘1,2,3,4-Butanetetracarboxylic acid (BTCA) powder (10 mM) 86 and sodium hypophosphite (SHP) monohydrate (5 mM) were added to cross-linked samples’

I guess that the BTCA and SHP were added before sonication and casting but the information about adding that ingredients appears after mentioning the film casting. The sentence could be understood that BTCA and SHP were added to already crosslinked samples. This part needs revision and clarification. 

  1. Line 89: ‘Flocculation experimentally determined’ should be ‘Flocculation was experimentally determined’
  2. Line 110: ‘ll samples are coated on a 30μm corona treated biaxially-oriented polypropylene (BOPP)’

The Authors should explain to the Readers what is the reason for using BOPP as a supporting film since the CNC/CZ films were self-standing.

  1. Line 135: ‘at a 1.6 Hz line scan rate ms−1 scan speed’

It is unclear what scanning rate was depicted by ms-1?

  1. Line 140: ‘Young's modulus was determined from the slope of the low strain’

The exact range of elastic deformation at which Young modulus was determined should be given.

  1. Line 145: ‘CNC/CZ /BTCA/SHP’

There is a space occurring in the denotation of crosslinked composite CNC/CZ /BTCA/SHP. Is it misspelling or it has some meaning?

  1. Line 156: ‘Samples of 10–20 mg’

In a typical thermogravimetric analysis of polymeric material the recommended sample mass is around 5 mg due to the limited heat transfer coefficient of polymers. The sample mass applied in this work is applicable in thermal analysis rather for coupled thermoanalytical methods. Moreover, the mass of samples in a series of measurement should vary by max. 10%. Here we have 100% mass difference. The differences in sample mass may cause shift in determined decomposition parameters. In this work  samples were in form of films with low thickness which could facilitate the heat transfer and thus the thermal inertia of the samples might be low. On the other hand, films demonstrated barrier properties that could slow down the evolution of gaseous products of decomposition and influence the measured mass loss. Some comment on the sample preparation for TGA measurement - grinding, cutting, size of pieces, would be informative for the readers.

  1. Line 151-153 This information has already been given in subchapter Self-standing films fabrication.
  2. Lines 153-160 The information on BTCA and SHP suits better the introduction part and actually have been missing there while reading.
  3. Line 179: ‘In summary, the addition of the BTCA cross linker to nanocomposite CNC-CZ 179 films generated films with improved transmittance at 500 nm.’

The description is not precise. The films with 30:1 and 10:1 ratio have better transmittance at 500 nm without crosslinking system.

  1. Figure 3: The axes should be described properly with the parameter name, not only the unit.
  2. Line191: The passage ‘(Figures 4, 5 lower values suggest better water resistance).’ should be reformulated into a regular sentence.
  3. Line 212: ‘The water absorption capacity of the CNC/CZ nanocomposite films was lower’

Does this passage refer to crosslinked CNC/CZ nanocomposites?

Line 215: There is ‘WVP value, 2.03±2.8’ should be ‘WVP value of 2.03±2.8’ and further ‘value of 4.4 ±0.8%’.

  1. Line 60: ‘surface characterized by SEM and AFM’

in this context – ‘surface, as characterized by SEM and AFM’.

  1. 4 Sample denotation varies from graph to graph, text formatting (font type and size, units in branches or not) in all Figures is inconsistent and needs to be unified. The mark 'b' is missing on the second graph.
  2. Line 241: The phrase ‘which makes them less susceptible to water absorption and a higher contact angle’ need reformulation.
  3. Line 234: ‘ This might suggest that the CZ protein changes how the CNC arrange themselves in the surface of the film.’

The statement needs further explanation with reference to literature or support with experimental results. The presentation of film morphology analysis prior to other experimental results would clarify this doubts.

  1. Line 252: ‘WVTR’ – misspelling or the abbreviation not explained.
  2. Line 263: ‘and 12 [cc/m²·day] for the 10:1 CNC/CZ ratio’

From the graph one can find that the OTR for 10:1 CNC/CZ is slightly over 50 not 12 [cc/(m2day)], as described in the text.

  1. Line 265: ‘The increase in OTR with CNC concentration is associated with the hydrophilic nature of CNC.’

This conclusion is not supported by experimental results and not discussed with respect of literature data on permeability of CNC- containing films. The cellulose nanocrystals itself are not permeable for gases so it is unclear why the increase of CNC content should decreased permeability. The reference cited in the next sentence describes not CNC system but CZ containing system, so is not relevant in discussion on the influence of CNC concentration on permeability. The porosity of the films was not evaluated in the study. The Authors mention observed porosity in the morphological studies but do not refer to morphological observations in the discussion of barrier properties. The morphology should be presented before the barrier properties discussion.

  1. Numbering of Fig 8a and 8b. Denotations a and b are not present in the Fig 8. They should be counted as separate figures.
  2. Morphology description: In the SEM photomicrographs of un-cross linked composites I see grain-like structure or some elongated domains resembling those observed in spinodal decomposition phase separation. The CNC are in general hydrophilic and CZ are considered hydrophobic. I suggest consideration of possible miscibility improvement of CNC and CZ by crosslinking.
  3. Is seems that CZ is forming grain like domains in un-cross linked composite materials. The photomicrographs of corn zein solution supplied to the same preparative procedure as composite materials would help to understand the morphology of composite materials.
  4. Line 297: ‘it was noted that CNC particles self-assembled in an orderly arrangement.’

Was it generally observed for all materials or seen in a particular type of material?

  1. Line 299: ‘formation of needle- like shape structure’

AFM scans should be included to illustrate the described morphology. The AFM phase mode would be very informative on the possible phase separation and the presence of CZ-rich domains.

  1. Line 304: the agglomeration of CZ and embedded in the CNC matrix’.

unclear phrase.

The Authors should provide to the Readers the information what is the typical behaviour and morphology of CZ upon drying from solution based on experiments or with respect to the relevant literature.

  1. Line 330: ‘significantly higher in the 330 crosslinked films except in CNC /CZ 6: 1 and 3: 1 ratios’

CNCs could be considered as the component capable of effective stress transfer and the effective amount of CNC was decreasing in the composites with increasing CZ content.

  1. Line 346: ‘both’

both of what?

  1. Line 356: ‘it seems that the cross linker had no significant effect on the thermal properties’

Well, if one compare the onset temperature of the first decomposition step, it seems that the less stable materials in the series of un-cross linked films (CNC:CZ 30:1 and 15:1) become the most stable after crosslinking. The table with determined parameters of thermal degradation, such as initial temperature of degradation (determined as Tonset), temperatures at the maximum rate of mass loss (Tmax) in the first decomposition step would allow quantitative evaluation of the effect of CZ addition on the film thermal stability and illustrate the difference between the crosslinked and un-cross linked samples. It seems that Tmax of crosslinked CNC:CZ 30:1 was by over 30oC higher than that for corresponding crosslinked CNC film.

  1. Line 368: ‘showed above average flexibility’

average of what? Perhaps some comparison of current results with literature data for other biofilms or films commonly used in packaging industry would be added to the subchapter devoted to mechanical properties.

  1. Line 375: “a Sheaf of wheat”

Why using upper case letter in the name of the structure?

  1. Line 379: ‘causes changes in both the surface and bulk of the films’

The fracturing and analysis of morphology in the bulk is mentioned in the experimental and conclusions, but it seems that bulk morphology was not discussed in the paper.

  1. Line 450: Some other paper is under the given link.
  2. Some reference have doi number some does not have it.

Extensive text editing is necessary in the whole manuscript (spelling).

Author Response

Thank you very much for your comments.

Reviewer 3 Report

Gaska et al. reported the manuscript entitled ‘‘Cellulose nanocrystals and corn zein oxygen and water vapor barrier biocomposite films.’’ This manuscript needs major revision before publication. Some comments are as follow:

  1. The author should explain why CNC used as a matrix in this work? Generally, CNC is used as a nanofiller in biocomposite film studies.
  2. Please mention the novelty of the current study.
  3. After loading of corn zein Young’s modulus decreased compared with CNC film. So what is the advantage of the filler? Packaging materials required higher mechanical properties.
  4. In lines 64 and 66, check the spelling errors (Engel and characteris). It should be angle…….careless writing.
  5. If possible the authors need to provide contact angle images of fabricated film samples.
  6. In line 111, Change from 30μm to 30 μm maintain consistency in the whole manuscript.
  7. Please rearrange Fig. 3 in a better way (X-axis from 190-1100 nm and clarity should be improved).
  8. For SEM images, it is suggested to use an arrow mark for the location of corn zein in the matrix.
  9. In Fig. 4, the author should explain why 1:15 showed lower water absorption than 1:30. Other samples are not in a trend, need more explanations.
  10. Regarding water vapor permeation results are not in a trend?
  11. Please calculate the thermal degradation temperature and mention it in a Table (hard to identify in the figure).

Author Response

Thank you very much for your comments.

Round 2

Reviewer 1 Report

The authors significantly improved and now warrants publication in Nanomaterials. 

Author Response

Dear reviewer, 

Thank you very much for your second review of the manuscript.

The work has been further revised in regards to grammar and the figure numbers have been corrected.

Best regards,

Shylee Belsey

Reviewer 2 Report

Dear Editor, Dear Authors,

The work is has been amended and can be published after a minor corrections.

The difficulty in checking of the revised paper was lack of clear indication of the changes made by Authors by referring to the line numbers and/or marking the changed text by a distinct font colour neither in the revised manuscript nor the Cover Letter.  

The manuscript requires careful proof reading to correct numerous misspellings, misplaced spaces and punctuation marks.

Furthermore, the following minor corrections are required:

  1. Line 10: There is: ‘Cellulose Nanocrystals (CNC) is’ should be ‘Cellulose Nanocrystals (CNC) are’
  2. Table 1: The mark ‘%’ by each given value is excessive since the unit is given in the table heading.
  3. Line 232: There is ‘water absorbance’ should be ‘water absorption’.
  4. Line 284: Regarding point 31 in the Cover Letter - Authors agreed in the response that the OTR values given in line 285 (of revised paper) refer to the crosslinked systems, however the sentence in the revised manuscript was not accordingly reformulated. The given values still seem to refer to uncrosllinked films. Please clarify it.
  5. Figure numbering in the text and captions requires correction. Authors renumbered the Fig. 8b into Fig. 9 but the numbering of the following figures and text passages referring to the figures were not adjusted. For example, in line 304 and 306 the text refers to the previous numbering of the Figures, so the reader is referred to the Fig. 8b which is now Fig. 9. There are two different figures with number 9, etc.
  6. Caption for table 2: I believe the caption for Table 2 should be: Temperatures at maximum rate of mass loss in the first degradation step of crosslinked and non-crosslinked films.
  7. In reference 13 there are missing bibliographic data (pages, volume)
  8. Reference 18 has wrong doi number.

Reviewer 3 Report

The authors have addressed all the major problems. So I recommend it for publication. Please check Figure 9 mentioned two times. 

Author Response

(The authors gave the same response as above.)
